# Advanced Thermoelectric Performance of SWCNT Films by Mixing Two Types of SWCNTs with Different Structural and Thermoelectric Properties

**DOI:** 10.3390/ma18010188

**Published:** 2025-01-04

**Authors:** Yutaro Okano, Hisatoshi Yamamoto, Koki Hoshino, Shugo Miyake, Masayuki Takashiri

**Affiliations:** 1Department of Materials Science, Tokai University, Hiratsuka 259-1292, Kanagawa, Japan; 3cajm010@mail.u-tokai.ac.jp (Y.O.); 3cajm057@mail.u-tokai.ac.jp (H.Y.); 3cajm049@mail.u-tokai.ac.jp (K.H.); 2Department of Mechanical Engineering, Setsunan University, Neyagawa 572-8508, Osaka, Japan; shugo.miyake@setsunan.ac.jp

**Keywords:** SWCNTs, thermoelectric, thermal conductivity, Seebeck coefficient, power factor, dimensionless figure-of-merit

## Abstract

Semiconducting single-walled carbon nanotubes (SWCNTs) are significantly attractive for thermoelectric generators (TEGs), which convert thermal energy into electricity via the Seebeck effect. This is because the characteristics of semiconducting SWCNTs are perfectly suited for TEGs as self-contained power sources for sensors on the Internet of Things (IoT). However, the thermoelectric performances of the SWCNTs should be further improved by using the power sources. The ideal SWCNTs have a high electrical conductivity and Seebeck coefficient while having a low thermal conductivity, but it is challenging to balance everything. In this study, to improve the thermoelectric performance, we combined two types of SWCNTs: one with a high electrical conductivity (Tuball 01RW03, OCSiAl) and the other with a high Seebeck coefficient and low thermal conductivity (ZEONANO SG101, ZEON). The SWCNT inks were prepared by mixing two types of SWCNTs using ultrasonic dispersion while varying the mixing ratios, and *p*-type SWCNT films were prepared using vacuum filtration. The highest dimensionless figure-of-merit of 1.1 × 10^−3^ was exhibited at approximately 300 K when the SWCNT film contained the SWCNT 75% of SWCNT (ZEONANO SG101) and 25% of SWCNT (Tuball 01RW03). This simple process will contribute to the prevalent use of SWCNT-TEG as a power source for IoT sensors.

## 1. Introduction

Since their discovery in the 1990s, carbon nanotubes (CNTs) have been used in various industrial applications such as rechargeable batteries, automotive parts, and sporting goods [1,2]. CNTs are tubular structures composed only of carbon atoms, and there are mainly two types: single-walled carbon nanotubes (SWCNTs), which are composed of a single tube layer, and multi-walled carbon nanotubes (MWCNTs), which consist of multiple layers. MWCNTs have a tube diameter of approximately 4–150 nm and are characterized by their high strength. They are expected to be used as a conductive additive in lithium-ion batteries for mobile devices and electric vehicles [3,4,5]. SWCNTs have a tube diameter of approximately 1–5 nm and are featured for their flexibility as well as excellent electrical and thermal conductivity. Taking advantage of these characteristics, single-walled CNTs are used as composite materials with rubber, resin, metal, etc., because of the increase in the strength [6,7,8]. Furthermore, SWCNTs show metallic or semiconducting characteristics depending on their structure, which is defined by their chiral index (*n*, *m*) [9]. In particular, semiconducting SWCNTs are increasingly being used in a wide variety of electronic devices, including transistors, memories, and sensors [10,11,12,13]. The use of semiconducting SWCNTs in electronic devices will enable higher speeds and lower power consumption.

Thermoelectric generators (TEGs) that transform thermal energy into electricity via the Seebeck effect are attractive electronic devices using semiconducting SWCNTs [14,15,16,17,18,19,20,21]. This is because the characteristics of semiconducting SWCNTs are perfectly suited for TEGs as standalone power sources for sensors on the Internet of Things (IoT) [22,23,24,25]. However, the thermoelectric performances of the SWCNTs are inferior to those of traditional inorganic materials [26]. The primary thermoelectric performance is the dimensionless figure-of-merit, *ZT*, which is expressed as *ZT* = *σS*^2^*T*/*κ*, where *σ*, *S*, *T*, and *κ* are the electrical conductivity, Seebeck coefficient, absolute temperature, and thermal conductivity, respectively. In addition, the power factor, *PF*, which is expressed as *PF* = *σS*^2^, is also an important factor of the thermoelectric performance. Therefore, *ZT* can be improved by reducing the thermal conductivity without decreasing the power factor, or by increasing the power factor without increasing the thermal conductivity.

To enhance the thermoelectric performance of SWCNTs, it is essential to understand the relationship between each thermoelectric parameter and the characteristics of the SWCNTs. The electrical conductivity mainly depends on the defects in the SWCNTs [27,28]. The SWCNTs exhibit a high electrical conductivity at a low defect density. The Seebeck coefficient mainly depends on the diameter and chirality of the SWCNTs. The SWCNTs exhibit a high Seebeck coefficient at a small diameter [29]. In the chiral index (*n*, *m*), when *n*-*m* is a multiple of 3, the SWCNTs have almost zero of the Seebeck coefficient due to obtaining metallic properties; otherwise, they have a relatively large Seebeck coefficient due to obtaining semiconducting properties. In general, the semiconducting SWCNTs exhibited a *p*-type Seebeck coefficient in the air because the adsorption of oxygen molecules onto the SWCNTs causes electron transfer from the SWCNTs to the oxygen molecules [30,31,32,33]. The thermal conductivity mainly depends on the defect and surface conditions of the SWCNTs [34,35,36,37]. The SWCNTs exhibit a low thermal conductivity at a high defect density. In addition, the thermal conductivity becomes low when the SWCNTs have rough surfaces, because the thermal resistance between the SWCNTs increases due to the decrease in the contact areas. In general, it is extremely difficult for SWCNTs to exist as individual SWCNTs, because they aggregate and form bundle structures due to van der Waals forces. However, the bundle size can be made as thin as possible by adjusting the ultrasonic dispersion conditions and adding surfactants to SWCNT inks.

Although the conditions of SWCNTs to achieve the best thermoelectric performance have been clarified, it is significantly difficult to fabricate the ideal SWCNTs with a high electrical conductivity and high Seebeck coefficient, while having a low thermal conductivity. Under these circumstances, attempts have been made to fabricate mixtures of SWCNTs and inorganic materials such as Bi_2_Te_3_ nanoplates to achieve as close as possible to the ideal thermoelectric performance [38,39]. Although the thermoelectric performance can be improved by mixing Bi_2_Te_3_, the toxicity and scarcity of tellurium (Te) and the uneven distribution of bismuth (Bi) resources are problematic. Therefore, to maximize the thermoelectric performance using only SWCNTs, one approach is to mix several types of SWCNTs with different structures and thermoelectric properties. There are several methods for synthesizing SWCNTs, and the structure and thermoelectric properties of SWCNTs vary depending on the synthesis method [40,41,42,43,44]. For instance, SWCNTs synthesized by the super-growth method, which are thick (3–5 nm in diameter) and contain many defects, have a relatively high Seebeck coefficient and low thermal conductivity [45]. In contrast, SWCNTs synthesized by the HiPco process, which are thin (approximately 1 nm in diameter) and contain few defects, have a relatively high electrical conductivity [46].

In this study, we used two types of SWCNTs with significantly different structural and thermoelectric properties: one with a large diameter and many defects, and the other with a small diameter and few defects. By mixing two types of SWCNTs by changing the mass ratio, several types of SWCNT inks were prepared using ultrasonic dispersion, followed by forming the *p*-type SWCNT films using vacuum filtration. The physical and thermoelectric characteristics of the SWCNT films were analyzed. As a result, the two types of SWCNTs were bundled and entangled with each other and did not mix with each other to form bundles. We maximized the *ZT* by an optimal mixing ratio of two types of SWCNTs. The method in this study is simple and therefore effective in improving the performance of SWCNT films towards TEGs as self-contained power sources.

## 2. Experimental Method

The manufacturing process of SWCNT inks and films is shown in Figure 1. As starting materials, we prepared two types of SWCNTs as shown in Table 1. These two types of SWCNTs have different tube diameters and lengths. In the following, the ZEONANO SG101 and Tuball 01RW03 will be referred to as SWCNT-SG and SWCNT-Tu, respectively. The two types of SWCNTs were combined with ethanol (Fujifilm Wako Pure Chemical, Osaka, Japan) to prepare the SWCNT inks. The total amount of SWCNTs was maintained at 80 mg while the mass ratio of SWCNT-SG and SWCNT-Tu was varied. To increase the uniformity of the inks, an ultrasonic homogenizer (Branson Sonifier SFX 250, Emerson, St. Louis, MO, USA) was used with an operating amplitude of 70% (maximum power: 200 W) for 60 min in cold water. The fabricated SWCNT inks were then used to prepare the SWCNT films by vacuum filtration. In this film preparation, 10 mL of the SWCNT ink was aspirated with a piece-loaded pipette and dropped uniformly into a 90 mm diameter mesh filter (PTFE, ADVANTEC, Tokyo, Japan) placed on a mesh holder in an aspiration bottle. This process was redone four times to produce a SWCNT film with a diameter of 80 mm and a thickness of 50–100 μm, according to the mixing ratio of the two types of SWCNTs. After 24 h of drying, the SWCNT film was peeled off the mesh filter.

The nanostructures of the two types of SWCNT powders were analyzed using field-emission transmission electron microscopy (FE-TEM; JEM-2100F, JEOL, Akishima, Japan) at an accelerating voltage of 200 kV. The surface morphologies in the SWCNT films were examined via field-emission scanning electron microscopy (FE-SEM; S-4800, Hitachi, Tokyo, Japan). The crystal structure of the films was examined via Raman microscopy, using a laser wavelength of 515 nm (XploRA, Horiba, Kyoto, Japan), and X-ray diffraction (XRD; Rigaku MiniFlex 600, Akishima, Japan) with Cu-*Kα* radiation (*λ* = 0.154 nm with 2*θ* ranging from 10° to 60°).

The thermal conductivity *κ* was determined according to the formula below: *κ* = *ρCD*, where *ρ*, *C*, and *D* are mass density, specific heat, and thermal diffusivity, respectively. The mass density measurement was performed using the Archimedes method using an analytical balance (AP225WD, Shimadzu, Kyoto, Japan) with ethanol as the dipping solvent. The specific heat measurement was performed by differential scanning calorimetry (DSC-60 PLUS, Shimadzu, Kyoto, Japan). The thermal diffusivity measurement was performed by thermometry based on non-contact laser-spot periodic heating radiation (TA33 thermowave analyzer, Bethel, Ishioka, Japan) with a precision within 5% [47].

The measurement of the Seebeck coefficient *S* of the SWCNT films in the in-plane direction was performed at approximately 300 K with a precision within 5% [39,48]. One edge of the film was attached to a copper plate and another edge was attached to an electric heater. The Seebeck coefficient was derived as the rate of the thermoelectromotive-force difference across the film to the temperature gradient measured between two K-type thermocouples with a 100 μm diameter attached to the film. The measurement of the Seebeck coefficient was redone eight times for each test piece, and then, the measured values were averaged. The measurement of the electrical conductivity *σ* in the in-plane direction was performed at approximately 300 K using a four-point probe method (RT-70V, Napson, Tokyo, Japan) with a precision within 3% [49]. The measurement of electrical conductivity was redone eight times for each test piece, and then the measured values were averaged. The in-plane power factor *σS*^2^ was derived from the determined Seebeck coefficient and the electrical conductivity. The dimensionless figure-of-merit *ZT* was derived from the determined power factor and thermal conductivity.

## 3. Results and Discussion

Figure 2 shows the nanostructure of the two types of SWCNT powders observed using FE-TEM. In Figure 2a, SWCNT-SG are mostly single-walled CNTs but also contain multi-walled CNTs. The diameter of single-walled CNTs is approximately in the range of 3–5 nm, and the typical diameter is 3.3 nm. Because the surface of the single-walled CNTs was relatively rough, the resulting bundles had gaps and a mesh-like structure in some places. In Figure 2b, SWCNT-Tu most perfectly contains single-walled CNTs with an approximate diameter of 1.2 nm, which is smaller than that of the SWCNT-SG. Because the surface of the single-walled CNTs was relatively smooth, single-walled CNTs are aligned and in close contact with each other. Therefore, the diameter and surface condition of the SWCNTs varied according to the difference in the synthesis method.

The surface morphology and microstructure of the SWCNT films mixing two types of SWCNTs are shown in Figure 3. The SWCNT film, comprising solely SWCNT-SG, is characterized by the presence of non-aligned, rough surface bundles with an approximate diameter of 100 nm (Figure 3a). When the mixing ratio, SWCNT-Tu/(SWCNT-Tu + SWCNT-SG), was 25%, individual SWCNTs were entangled in bundles, indicating that the two types of SWCNTs are not intermingled to form bundles (Figure 3b). Increasing the mixing ratio did not change the tendency to form individual bundles, while the number of bundles of SWCNT-Tu increased (Figure 3c,d). In Figure 3e, the SWCNT film, comprising solely SWCNT-Tu, is characterized by the presence of aligned bundles with smooth surfaces. The diameter of the bundles, which was an approximate diameter of 300 nm, was larger than that of the SWCNT-SG bundles shown in Figure 3a.

Figure 4 shows the Raman spectra of the SWCNT films prepared using SWCNT inks with different mixing ratios of two types of SWCNTs. Note that Raman spectra of SWCNT films are averaged signals of SWCNT-SG and SWCNT-Tu. In Figure 4a, the Raman spectra of all SWCNT films exhibited *G*- and *D*-bands at approximately 1590 and 1350 cm^−1^, respectively. The *G*-band is a spectrum derived from graphite, which consists of hexagonal lattice carbon atoms, and the *D*-band appears when the crystal lattice of SWCNT contains disturbances in the carbon basal plane lattice (impurities, edges, defects, etc.). Therefore, the ratio of the integrated intensities of the *G*-band and the *D*-band is an index of the crystallinity of the SWCNTs. Figure 4b shows the intensity ratio (*I_G_/I_D_*) of the SWCNT films. The *I_G_/I_D_* ratio of the SWCNT film, comprising solely SWCNT-SG, was determined to be 2.6. The *I_G_/I_D_* ratio increased linearly as the amount of SWCNT-Tu added increased. In the SWCNT film made solely from SWCNT-Tu, the *I_G_/I_D_* ratio reached 33.7. Therefore, the structural defects in the SWCNT films decreased when increasing the loading ratio of SWCNT-Tu. This phenomenon is reflected in the surface morphologies of the SWCNT films. SWCNTs with rough surfaces exhibit a high density of defects, whereas SWCNTs with smooth surfaces display a low density of defects. In addition, the crystal structures determined from the X-ray diffraction patterns of the SWCNT films are presented in the Appendix A.

Figure 5 shows the thermal and physical properties of SWCNT films prepared using SWCNT inks with different mixing ratios of two types of SWCNTs. In Figure 5a, the in-plane thermal diffusivity increased linearly as the addition ratio of SWCNT-Tu increased. The thermal diffusivity of the SWCNT film comprising solely SWCNT-Tu, which was 54.4 mm^2^/s, was three times higher than that of the film comprising solely SWCNT-SG. This trend occurred because SWCNT-Tu had fewer defects and thermal resistance between the SWCNTs compared to those of SWCNT-SG. In Figure 5b,c, the mass density and specific heat slightly increased as the addition ratio of SWCNT-Tu increased. As a result, the in-plane thermal conductivity was derived from thermal diffusivity, mass density, and specific heat, as shown in Figure 5d. The thermal conductivity of the SWCNT film comprising solely SWCNT-SG was 5.9 W/(m·K). The thermal conductivity increased linearly as the addition ratio of SWCNT-Tu increased. The thermal conductivity of the SWCNT film comprising solely SWCNT-Tu was 27.4 W/(m·K), which was 4.6 times higher than that of the SWCNT film comprising solely SWCNT-SG. Therefore, to achieve a high *ZT*, it is desirable to have as low a thermal conductivity as possible, and the SWCNT film comprising solely SWCNT-SG is best suited for this condition. For an individual SWCNT, the thermal conductivity has been empirically determined to range from 500 to 7000 W/(m·K) near 300 K [50,51,52]. Thus, the thermal conductivity is drastically reduced by the formation of bundles and then films, even though the SWCNTs, such as SWCNT-Tu, have smooth surfaces and are firmly bonded to each other. As the surface of the SWCNT becomes rougher, such as SWCNT-SG, the thermal resistance between the SWCNTs increases, further reducing the thermal conductivity.

The thermoelectric properties in the in-plane direction of the SWCNT films prepared using SWCNT inks with different mixing ratios of two types of SWCNTs are shown in Figure 6. In Figure 6a, the electrical conductivity of the SWCNT film comprising solely SWCNT-SG was 26.8 S/cm. The electrical conductivity greatly increased as the addition ratio of SWCNT-Tu increased. The electrical conductivity of the SWCNT film comprising solely SWCNT-Tu was 1230 S/cm. The electrical conductivity of the individual SWCNTs has been reported to be in the range of 10^4^–10^5^ S/cm near 300 K [53,54]. Thus, the formation of bundles and then films drastically reduces the electrical conductivity, even though the SWCNTs, such as SWCNT-Tu, have smooth surfaces and are firmly bonded to each other. Furthermore, the electrical conductivity of the SWCNT film solely comprising only SWCNT-SG is further reduced to 1/46 of that of the SWCNT film solely comprising SWCNT-Tu. There is a large difference between the increased rates of thermal and electrical conductivities when the addition ratio of SWCNT-Tu is increased. The exact mechanism behind this difference is unclear, but there are two possible explanations. The first explanation is that SWCNT-SG has a high defect density, as shown in Figure 4b, and numerous dangling bonds are generated on the SWCNT surface, which hindered the carrier flow because the carriers were trapped by the dangling bonds. Heat flow is also affected by defects (dangling bonds) on the SWCNT surface, but the effect of defects is less than the carrier flow because heat flow is not subject to electrical forces. The second explanation is that SWCNT-SG has a large surface area due to the rough surface and gaps between the SWCNTs as shown in Figure 2a, which causes more oxygen molecules to be adsorbed on the SWCNT surface. As a result, the carrier flow is impeded by the oxygen molecule layers as an electrical resistive layer. The oxygen molecule layers also act as a thermal resistance, but it has less effect than the carrier flow because the heat flow also passes through the electrical resistive layer.

In Figure 6b, the Seebeck coefficient of the SWCNT film comprising solely SWCNT-SG was 43.2 μV/K. When the addition ratio of SWCNT-Tu increased, the Seebeck coefficient decreased. The SWCNT film comprising solely SWCNT-Tu was 26.9 μV/K. This trend is related to the difference in carrier concentration of two types of SWCNTs. The relationship between *S* and carrier concentration *n* is expressed by Equation (1).
(1)$$S=\frac{8\pi^{2}{k_{B}}^{2}}{3eh^{2}}m^{*}T{\left(\frac{\pi }{3n}\right)}^{\frac{2}{3}}$$
where *k_B_*, *h*, *m**, and *T* are the Boltzmann constant, Planck’s constant, effective mass, and absolute temperature, respectively. The Seebeck coefficient was inversely proportional to the carrier concentration assuming that the effective mass remains unchanged. Since the Seebeck coefficient of the SWCNT film comprising solely SWCNT-Tu was lower than that of the SWCNT film comprising solely SWCNT-SG, the carrier concentration of SWCNT-Tu is higher than that of SWCNT-SG. This is probably because SWCNT-Tu has a higher content of metallic SWCNTs than that of SWCNT-SG, and this effect is thought to be greater than the effect of increasing the Seebeck coefficient due to the smaller diameter of SWCNT-Tu compared to SWCNT-SG [29].

In Figure 6c, the power factor of the SWCNT film comprising solely SWCNT-SG was 5.0 μW/(m·K^2^). The electrical conductivity greatly increased as the addition ratio of SWCNT-Tu increased. The power factor of the SWCNT film comprising solely SWCNT-Tu was 89 μW/(m·K^2^), which was 18 times higher than that of the SWCNT film comprising solely SWCNT-SG. Therefore, to achieve a high *ZT*, it is desirable to have as high a power factor as possible, and the SWCNT film comprising solely SWCNT-Tu is best suited for this condition, which is contrary to the case of thermal conductivity.

In Figure 6d, the *ZT* of the SWCNT film comprising solely SWCNT-SG was 2.5 × 10^−4^. As the addition ratio of SWCNT-Tu was enhanced, the *ZT* increased, and the highest *ZT* of 1.1 × 10^−3^ was obtained at an addition ratio of 25% due to the best balance between the power factor and thermal conductivity. At an addition ratio of 50%, the *ZT* once decreased, but the *ZT* increased again when further increasing the addition ratio due to the contribution of the high power factor. As a result, the *ZT* of the SWCNT film at an addition ratio of SWCNT-Tu of 25% was 4.5 times and 1.2 times higher than the SWCNT films comprising solely SWCNT-SG and SWCNT-Tu, respectively. Therefore, we demonstrated that the *ZT* can be maximized by optimizing the mixing ratio of the two types of SWCNTs. Since the fabrication process in this study is straightforward, it contributes to the widespread use of SWCNT thermoelectric generators with advanced performance as power sources for IoT sensors. However, the thermoelectric performances of SWCNT films are still lower than those of established inorganic thermoelectric materials, as shown in the Appendix A. Therefore, the next challenge is to further improve the performance of SWCNTs through approaches such as the complete mixing of two or more SWCNTs using surfactants.

Finally, to investigate the long-term stability of the SWCNT films, we measured their thermoelectric properties again 250 days after the film preparation, and the results are shown in the Appendix A. The Seebeck coefficient and electrical conductivity of the SWCNT films did not show any significant degradation at all the mixing ratios. Therefore, we conclude that the SWCNT films in this study have long-term stability in their thermoelectric properties.

## 4. Conclusions

To maximize the thermoelectric performance of *p*-type SWCNT films, two types of SWCNTs, which are SWCNT-SG and SWCNT-Tu, were mixed to produce the SWCNT inks. Sequentially, the SWCNT films were fabricated by vacuum filtration. According to the FE-TEM observations, SWCNT-SG had rough surfaces with an approximate diameter of 3–5 nm and gaps between the SWCNTs, while SWCNT-Tu had smooth surfaces with an approximate diameter of 1 nm and no gaps between the SWCNTs. When the addition ratio of SWCNT-Tu was increased, the power factor increased, but the thermal conductivity also increased. The best performance, a *ZT* of 1.1 × 10^−3^ at approximately 300 K, was obtained when the addition ratio of SWCNT-Tu was 25%. Therefore, by mixing SWCNTs with different structures and thermoelectric properties, we were able to achieve high thermoelectric performances that could not be achieved with only one type of SWCNT. In addition, the SWCNT films in this study have long-term stability in their thermoelectric properties. Given this achievement, the developed SWCNT film will contribute to the advancement of SWCNT thermoelectric generators as power sources for IoT sensors.

## Figures and Tables

**Figure 1 materials-18-00188-f001:**
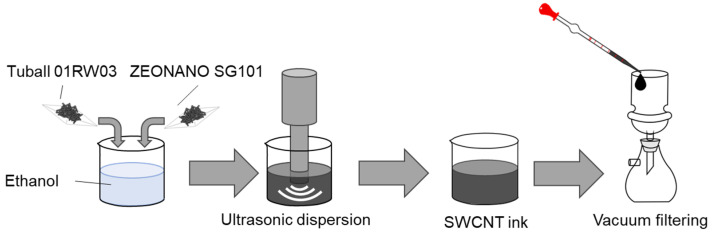
Manufacturing process of SWCNT inks and films.

**Figure 2 materials-18-00188-f002:**
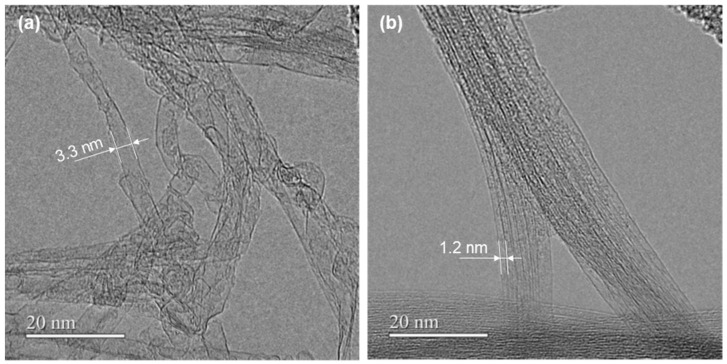
Nanostructure of SWCNTs determined by FE-TEM. (**a**) SWCNT-SG and (**b**) SWCNT-Tu.

**Figure 3 materials-18-00188-f003:**
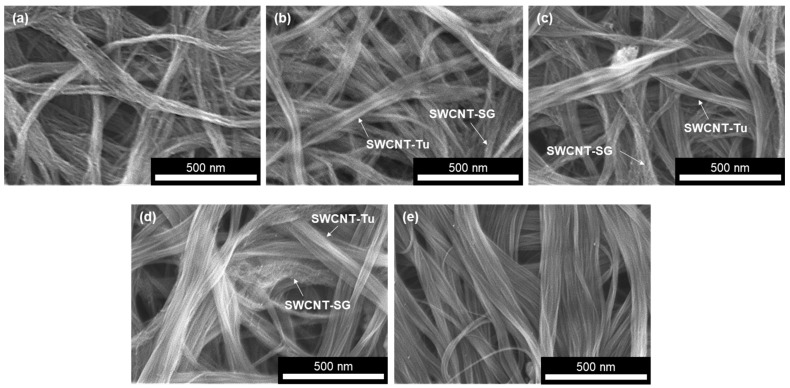
Microstructure and surface morphology of SWCNT films with different mixing ratios at SWCNT-Tu/(SWCNT-Tu + SWCNT-SG) of (**a**) 0%, (**b**) 25%, (**c**) 50%, (**d**) 75%, and (**e**) 100%.

**Figure 4 materials-18-00188-f004:**
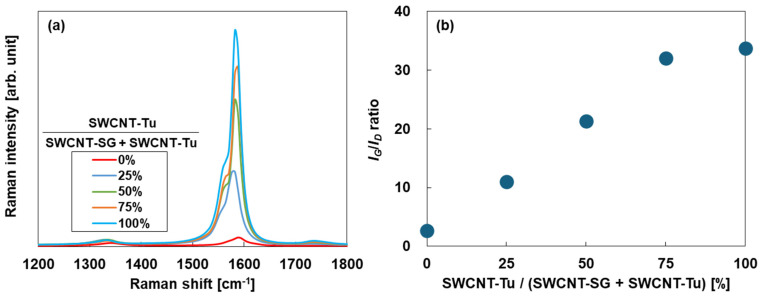
(**a**) Raman spectra of the SWCNT films with different mixing ratios of SWCNTs and (**b**) the relationship between the Raman intensity ratio (*I_G_*/*I_D_*) of the SWCNT films and the mixing ratio of SWCNTs.

**Figure 5 materials-18-00188-f005:**
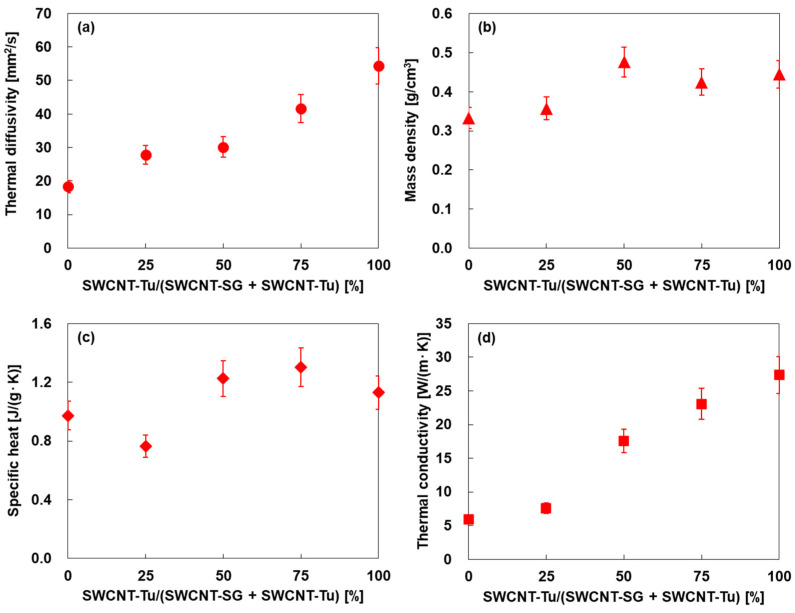
Thermal and physical properties of SWCNT films with different mixing ratios of SWCNTs. (**a**) In-plane thermal diffusivity, (**b**) mass density, (**c**) specific heat, and (**d**) in-plane thermal conductivity.

**Figure 6 materials-18-00188-f006:**
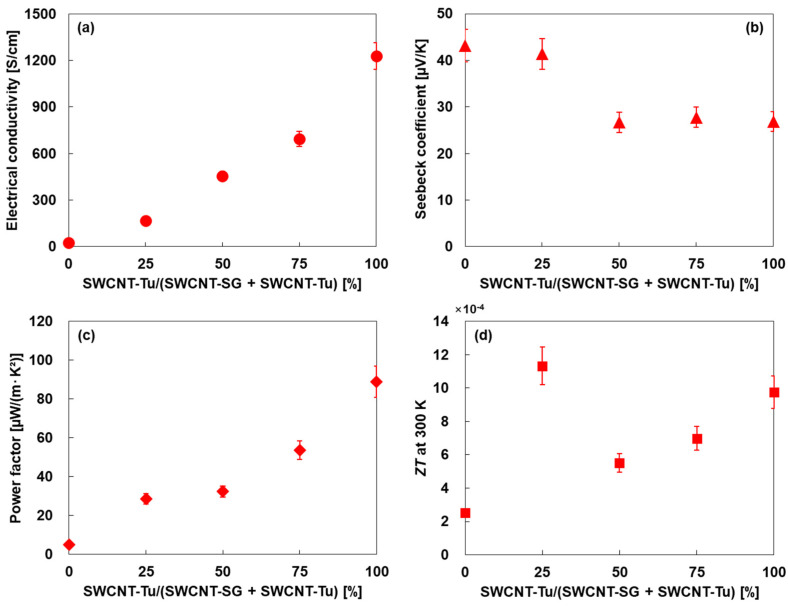
In-plane thermoelectric properties of SWCNT films with different mixing ratios of SWCNTs. (**a**) Electrical conductivity, (**b**) Seebeck coefficient, (**c**) power factor, and (**d**) dimensionless figure-of-merit.

**Table 1 materials-18-00188-t001:** Characteristics of SWCNTs.

SWCNT Name	Abbreviation Name	Tube Diameter	Tube Length	BET Surface Area	Growth Method	Manufacturer
ZEONANO SG101	SWCNT-SG	3–5 nm	350 μm	1200 m^2^/g	Super-growth	ZEON (Tokyo, Japan)
Tuball 01RW03	SWCNT-Tu	1.6 ± 0.4 nm	≤5 μm	1090 m^2^/g	Floating catalyst CVD	OCSiAl (Luxembourg)

## Data Availability

The raw data supporting the conclusions of this article will be made available by the authors on request. The data are not publicly available due to privacy.

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
