# Peer review of "Advanced Thermoelectric Performance of SWCNT Films by Mixing Two Types of SWCNTs with Different Structural and Thermoelectric Properties"

_materials, 2025, doi:10.3390/ma18010188_

Round 1

Reviewer 1 Report

Comments and Suggestions for Authors

The authors should address the following issues:

1.       On Line 32, the authors should highlight key industrial applications for CNTs.

2.       Line 40 still requires an explanation of why SWCNTs are composite with rubber, resin, metal, etc.

3.       Why are semiconducting SWCNTs increasingly being used in electrical devices such as transistors, memory, and sensors? Discuss it.

4.       On lines 68-70, may the authors have explained how to prevent the bundle form of the SWCNT structure?

5.       It would be preferable if the authors addressed the literature's Thermoelectric Performance/Properties-Based SWCNTs. Discuss their findings and issues. Then, explain why the current study is significant for researchers working on CNTs, as well as how the present research may solve those issues.

6.       The features and purpose of SWCNT films made by combining two types of SWCNTs should be addressed in the introduction.

7.       For Figure 2, which of the larger CNTs should be mentioned, and why?

8.       The authors may have explained why the thermal diffusivity of the SWCNT film containing just SWCNT-Tu is greater than that of the film containing only SWCNT-SG?

9.       To attain a high ZT, it is preferable to have low thermal conductivity. Why? Discuss it.

10.   The caption should be at the bottom of the figure. For example, Fig. 5's caption does not appear adjacent to the figure.

11.   The authors said that “SWCNT-SG has a large surface area due to the rough surface and gaps between the SWCNTs”. As a result, all CNTs should have their BET surface area calculated and explained.

12.   Is the Seebeck coefficient dependent on the diameter of the CNTs?

13.   It would be better if the authors discussed the XRD for the SWCNTs.

14.   It would be better if the authors included a proof of concept or application in Thermoelectric Performance using SWCNTs.

15.   Challenges of the present study should be discussed.

Reviewer 2 Report

Comments and Suggestions for Authors

Statistical analysis (e.g., standard deviation, error bars) should be provided to show the reproducibility and statistical significance of the results, especially the maximum ZT value reported.

A more extensive comparison of the obtained ZT values with those of other known thermoelectric materials would help place the results in a broader context and highlight the potential of this approach.

The long-term stability of the thermoelectric performance of the fabricated films needs to be evaluated. This is critical for practical applications, particularly in IoT devices.

Round 2

Reviewer 1 Report

Comments and Suggestions for Authors

The manuscript can be accepted for publication in its present form.